



# Convective gravity wave events during summer near 54°N, present in both AIRS and RMR Lidar observations

Eframir Franco-Diaz[1], Michael Gerding[1], Laura Holt[2], Irina Strelnikova[1], Robin Wing[1], Gerd Baumgarten[1], and Franz-Josef Lübken[1]

[1]Leibniz Institute for Atmospheric Physics at the Rostock University, Schloss-Str. 6, 18225 Kühlungsborn, Germany
[2]NorthWest Research Associates, 3380 Mitchell Lane, Boulder, CO 80301-2245, USA

**Correspondence:** Eframir Franco-Diaz (franco@iap-kborn.de)

**Abstract.** We connect tropospheric deep convective events over Western Europe, as measured by the 8.1 $\mu$m radiance observations from NASA's Aqua satellite's Atmospheric Infrared Sounder (AIRS), to horizontal brightness temperature variance in the 4 $\mu$m AIRS channel (maximum sensitivity at around 40 km) and temperature perturbations in vertical lidar profiles (between 33-43 km) over Kühlungsborn, Germany (54.12°N, 11.77°E). To account for wave propagation conditions from the
troposphere to the stratosphere, we also consider the horizontal winds in the troposphere and stratosphere using ECMWF. In this work, we highlight sporadic peaks in gravity wave activity in summer greatly exceeding those typical of summer, which is generally a season with lower wave activity compared to winter. Although these events are present in roughly half of the years (between 2003 and 2019), we focus our study on two case study years (2014 and 2015). These case study years were chosen because of the high cadence of lidar soundings close in time to the convective events. These events, while sporadic,
could contribute significantly to the zonal mean momentum budget and are not accounted for in weather and climate models.

## 1 Introduction

Earth's atmosphere is a complex system affected by dynamical processes over a broad range of spatial and temporal scales. Among the dynamic processes affecting the atmosphere are various waves spanning from planetary-scale Rossby and Kelvin waves, to synoptic-scale waves and all the way down to small-scale gravity waves. Gravity waves have spatial scales from
around a kilometre to thousands of kilometres and impact all layers of the atmosphere. They affect the dynamics, physics and chemistry of the atmosphere while also providing a coupling mechanism linking different regions of the atmosphere (Plougonven et al., 2020). Gravity waves are a fundamental driver of the circulation of the middle atmosphere, as they transport energy and momentum from the troposphere to higher altitudes. Since a large portion of the GW spectrum is under-resolved by climate and weather prediction models, their effects on the circulation must be parameterized (e.g., Alexander et al., 2010).
The main sources for gravity wave production are orography, convection, frontal systems and wind shear (e.g., Hoffmann et al., 2013; Alexander et al., 1995; Plougonven et al., 2003). In the middle atmosphere, unbalanced flows in the vicinity of jet streams, body forcing accompanying localized wave dissipation, wave–wave interactions, auroral heating and eclipse cooling are also known sources for gravity waves (Fritts and Alexander, 2003). In this study, we focus on convectively generated gravity





waves. Several studies have suggested that convective activity is at least as important as orographic sources for generating
gravity waves (e.g., Fritts and Nastrom, 1992; Hertzog et al., 2008; Plougonven et al., 2013; Jewtoukoff et al., 2015; Holt et al.,
2017, 2023). These studies have shown that while momentum fluxes are typically larger locally over orography, convective
gravity waves potentially contribute more to the zonal mean momentum flux because of their ubiquity. Unfortunately, the
properties of convective gravity waves and their impact on the circulation have been a challenge to characterize because of their
intermittent nature and large range of phase speeds, frequencies, and spatial scales. This means that despite their established
influence on the middle atmosphere circulation, the parameterizations that account for the effects of convective gravity waves
in climate and weather prediction models are not well constrained by observations. Constraints for parameterizations are
absolutely necessary because even with increases in computing power, climate models will be run at resolutions too coarse to
resolve the full gravity wave spectrum. In fact, even many cloud-resolving models down to 1 km horizontal grid spacing still
require gravity wave parameterizations because the waves and their effects are still underrepresented (e.g., Holt et al., 2016;
Stephan et al., 2019; Polichtchouk et al., 2022a, b). Studies of convective gravity waves are thus required to better constrain
physical parameterizations in models, which can improve forecast skill and accuracy from weather to climate (Marlton et al.,
2021).

As comprehensive global observations of convective gravity wave properties are not available, the use of observationally
validated cloud-resolving simulations is being explored as a tool for characterizing convective gravity waves. One of the
challenges of using cloud-resolving simulations for this purpose is that even with a full-physics cloud-resolving model, it is
still a challenge to accurately reproduce the locations, timing, and intensity of individual convective rain cells (Stephan and
Alexander, 2015). This makes it difficult to validate the convective gravity waves in these simulations. One method that has
been developed to address this issue is to force high-resolution simulations with observationally derived latent heating profiles
(Grimsdell et al., 2010; Stephan and Alexander, 2015; Stephan et al., 2016; Bramberger et al., 2022; Kruse et al., 2023).
For example, with this method, Stephan and Alexander (2015) found that both the gravity wave pattern and amplitudes were
accurately reproduced compared to AIRS observations. This method has significant potential to improve our understanding of
convective gravity waves, but this method also relies on high-quality observations of convective gravity waves to validate the
simulations.

Whether they are used to characterize wave properties or validate high-resolution simulations, our knowledge of convective
gravity waves can be improved by means of high-resolution measurements. These high-resolution observations can be obtained
by lidars (spatial scales of 150m on the vertical and temporal scales of 5 minutes (Chanin and Hauchecorne, 1981)), which
can measure almost every night, as long as good weather conditions prevail. New technological developments in lidars with
the capability to observe in daylight allow for higher cadence and better data coverage measurements, that enable us to better
separate gravity waves and tides, as well as better identify inertia gravity waves. (e.g., Baumgarten, 2010; Strelnikova et al.,
2021). Baumgarten et al. (2017) and Strelnikova et al. (2021) used lidar data from the daylight-capable Rayleigh-Mie-Raman
lidar located at Kühlungsborn, Germany to obtain gravity wave climatologies. They showed a distinct seasonal cycle in gravity
wave potential energy densities, with a maximum in winter. However, there was still significant gravity wave activity in the



summer months. In this study, we use the daylight-capable Rayleigh-Mie-Raman lidar located at Kühlungsborn, Germany in combination with the AIRS satellite to study gravity wave activity during the summer season in this region.

Hoffmann and Alexander (2010) optimized an existing method developed by Aumann et al. (2003) to detect deep convective clouds and correlate deep convection with stratospheric gravity waves using AIRS radiance measurements. Applying this method they concluded that the observed gravity waves near deep convective clouds during the summer over the United States exhibit concentric semicircular arc patterns that look remarkably similar to convective waves produced by models, however, they were unable to explicitly connect the wave to a point source. In later work, Hoffmann et al. (2013), identified between 4

and 18 stratospheric gravity wave hotspots, depending on the season, as significant convective and orographic sources. Most of the convective hotspots occurred in the summer season (May to August). In this study, we highlight significant gravity wave events over Kühlungsborn that were captured by both AIRS and the lidar. We adapt the method of Hoffmann et al. (2013) to identify stratospheric gravity wave events and deep convection in AIRS. We use this information in conjunction with high-resolution lidar profiles of gravity wave potential energy density (GWPED) to investigate the connection between

deep convection and stratospheric gravity wave activity over Kühlungsborn. In Section 2, we present a brief description of the instruments and datasets used. In Section 3, we describe the numerical methods applied to the data. In Section 4, we compare proxies for gravity wave activity computed from the lidar data and AIRS satellite data to convective activity estimated from AIRS. We also discuss the background wind conditions and observational filters. In Section 5 we provide a discussion of our results. Finally, we present our conclusions in Section 6.

## 75  2   Instruments and Datasets

### 2.1   AIRS Instrument

The Atmospheric Infrared Sounder (AIRS) is a nadir-viewing hyper-spectral infrared radiometer on board the AQUA satellite (Parkinson, 2003). AQUA is a polar-orbiting, sun-synchronous satellite orbiting at an altitude of 705 km with a 98° inclination. It achieves global coverage in 14.5 orbits (99-minute period), with 2 overpasses over Kühlungsborn each day (depending on

the way the swaths overlap, there could be more). The satellite was launched on May 4, 2002, and it has the mission to collect data on the Earth's water cycle including measurements of cloud properties, atmospheric temperature and humidity, as well as land and ocean skin temperatures (Aumann et al., 2003). We will take advantage of this cloud detection capability to identify summer periods where convective activity is observed over Europe.

The AIRS instrument scans the atmosphere across its path with a swath that is 1780 km wide. This scan width is composed

of 90 footprints that have a diameter of 13.5 km at nadir and increase in size off-nadir. The daytime crossing (ascending node, flying northward) occurs at 13:30 local time, while the nighttime crossing (descending node, flying southward) occurs at 01:30 local time. In this work, the nighttime scan (descending node) is used as it provides lower noise levels and better vertical resolution as it neglects the effects of non-local thermodynamic equilibrium (non-LTE) (Hoffmann et al., 2013).

For this work, we use the special AIRS gravity wave product developed by Hoffmann (2021) using the level 1B infrared

radiances. The processing of the level 1B AIRS data to obtain this special gravity wave product is described in Hoffmann and





Alexander (2010) and Hoffmann et al. (2013). To estimate gravity wave activity in this work, we use the 4.3 $\mu$m brightness temperature variance and perturbation variables included in the AIRS data product. The brightness temperature anomalies in this product were obtained by fitting and subtracting a fourth-order polynomial to the cross-track radiances to remove the large-scale background as well as limb brightening effects. Hoffmann and Alexander (2010) concluded that the method is most

sensitive to gravity waves with horizontal wavelengths between about 50 and 1000 km and vertical wavelengths greater than $\sim$15 km. The method is also sensitive to altitudes between 20-65 km but is most sensitive to altitudes between 30-40 km since the average kernel function of the AIRS channels in the 4.3 $\mu$m (2322.6–2366.9 cm$^{-1}$) CO$_2$ emission band has a broad peak in this altitude range in the stratosphere Hoffmann and Alexander (2010).

To detect high cloud tops connected to large convective events, we use the 8.1 $\mu$m brightness temperatures also included in

the AIRS special gravity wave product. Due to Earth's infrared atmospheric window, which occurs in a region roughly between 8 and 14 $\mu$m, the atmosphere is essentially transparent in this part of the spectrum. This window is sometimes narrowed because of clouds and in areas of high humidity because of water vapour absorption. The window is dominated by surface emissions in cloudless conditions; however, when low cloud temperatures are present at higher altitudes in the troposphere and lower stratosphere, optically thick clouds can be identified by relatively lower brightness temperatures in the AIRS 8.1 $\mu$m emission.

As in Hoffmann and Alexander (2010), we use both the 8.1 $\mu$m and 4.3 $\mu$m brightness temperature to concurrently identify deep convection and gravity waves in the geographical area around Kühlungsborn.

## 2.2 RMR lidar

Lidars are the only high-resolution, ground-based remote sensing technique capable of measuring temperature from 15 to 90 km (e.g., Baumgarten, 2010). The Leibniz-Institute of Atmospheric Physics (IAP) located in Kühlungsborn, Germany (54°

N, 12° E) has been measuring temperatures in the stratosphere-lower mesosphere since 1997 using a Rayleigh-Mie-Raman (RMR) lidar systems. From 2009 to the present, it is home to one of only two daylight-capable lidars for this altitude range in the world currently in operation (Gerding et al., 2016).

To calculate the temperature profiles from the lidar soundings, the classical hydro-static integration method is used (Hauchecorne et al., 1991; Hauchecorne and Chanin, 1980). Some modifications to this technique are required to correct for additional optics

needed for daytime sounding. The details can be found in Gerding et al. (2016).

RMR lidar systems measure vertical temperature profiles from the ground to above 90 km, depending on the temporal integration used. To reduce the statistical uncertainty of our mesospheric temperature measurements, the lidar profiles have been integrated vertically at 1 km resolution and temporally at 2 hours with a 15-minute step.

## 2.3 ECMWF Operational Forecast Data

The Integrated Forecast System (IFS) of the European Center for Medium-Range Weather Forecast (ECMWF) is a global, hydrostatic numerical weather prediction model. In this work, we use horizontal winds from implementation cycle 46r1 of IFS. This product has a horizontal grid spacing of approximately 9 km on 137 pressure levels and is output every 6 hours. In this



study, we use this product to show the background winds between approximately 10 and 40 km. This allows us to understand the gravity wave propagation with respect to the background winds.

## 2.4 Spectral sensitivity of Lidar and AIRS

The AIRS and lidar measurements have very different observation filters, i.e., they are sensitive to different parts of the gravity wave spectrum. AIRS brightness temperatures provide a snapshot in time of the horizontal structure, but cannot provide information on the vertical wavelengths or intrinsic frequencies of gravity waves. Lidar provides near-continuous, high-resolution vertical profiles, but cannot provide information on the horizontal structure or intrinsic frequencies of gravity waves. Furthermore, the background temperature is removed from AIRS and lidar data in very different ways. As mentioned above, AIRS brightness temperature anomalies are obtained by removing a 4th order cross-track polynomial, which corresponds to ∼500 km. The lidar data is processed with spatial (vertical) and temporal Butterworth filters. Figure 1 shows the vertical wavelengths that are present after the vertical and temporal filters are applied to the lidar data in blue. It also includes the AIRS response function for vertical wavelengths (Hoffmann and Alexander, 2009). One key feature we point out is that the lidar vertically filtered data does not overlap with AIRS. This is important for this study, as we will show that the same convective gravity wave events can be observed by instruments with very different observational methodologies.

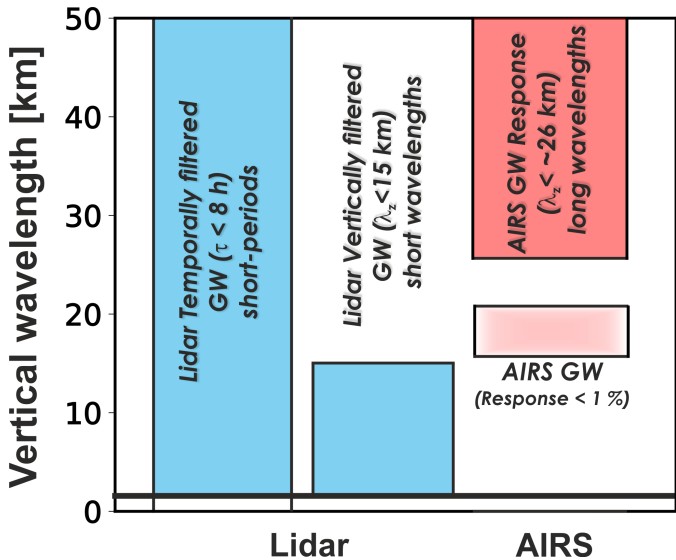

**Figure 1.** This sketch shows the vertical wavelengths that are covered by AIRS, as well as the temporally filtered and vertically filtered lidar data.





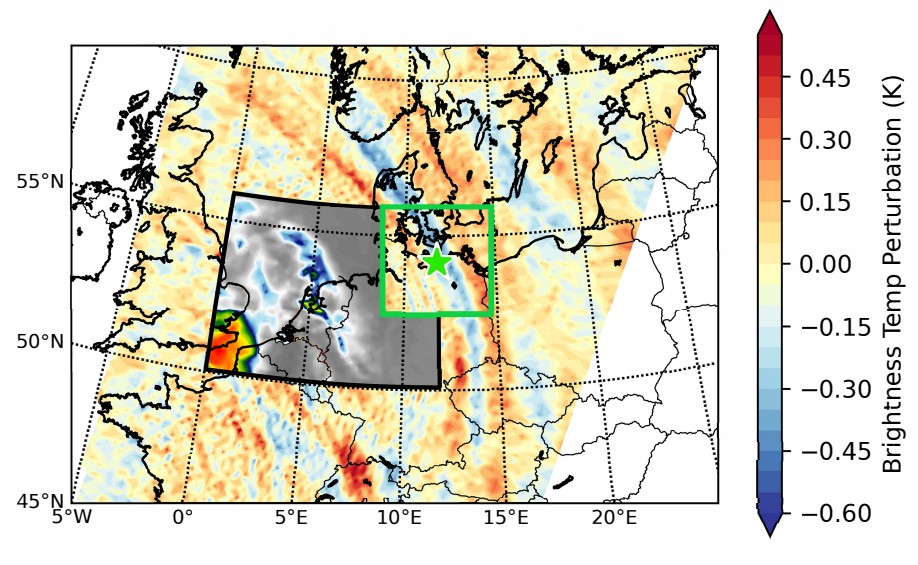

**Figure 2.** A single descending swath of AIRS brightness temperature perturbation data from the 4 $\mu$m channel is shown for the date of July 19, 2014 (blue, yellow and red). The green star shows the lidar location. The daily mean brightness temperature variance was calculated from the perturbation data for the area enclosed in the green box, and the results are shown in Figure 3. The black box shows the area taken for the 8 $\mu$m emission. The western half of the green box is also included in the area taken for the 8 $\mu$m emission.

## 3    Methodology

In this work, gravity wave events are identified by using the 4.3 $\mu$m brightness temperature variances in the AIRS gravity wave product Hoffmann (2021). We use the variance as a proxy for gravity wave activity. Figure 2 shows the AIRS 4.3 $\mu$m brightness temperature perturbation for July 19, 2014. As described in Section 2.1, AIRS is most sensitive to horizontal wavelengths between 50-1000 km and vertical wavelengths longer than ~15 km. This figure also shows the 8.1 $\mu$m AIRS brightness temperature inside the black rectangle. The green, yellow, orange, and red contours in the 8 $\mu$m data plot represent deep convective clouds. The 4.3 and 8.1 $\mu$m AIRS brightness temperatures together suggest that this event is a convective gravity wave with a brightness temperature amplitude of ~0.5 K. Figure 3 shows the brightness temperature variance as a function of time-averaged over an area of 400×400 km around Kühlungsborn (i.e. bounded by 52.35°N and 55.94°N in latitude and 8.68°E and 14.83°E in longitude as shown in the green box on Figure 2). The largest gravity wave activity is seen during the winter periods of both years, but there are also smaller local maxima observed during summer. We will make the case in subsequent sections that these peaks are caused by strong convective activity. It can be seen in Figure 3 that there are many data points in summer where the average brightness temperature exceeds a threshold of 0.02 K$^2$ (inside the brown box). We chose to focus on gravity wave events exceeding this threshold for this study based on the threshold used by Hoffmann et al. (2013) for brightness temperature variances at 60°N in June for descending node measurements. The days that exceed



the threshold are listed in Table 1 for 2003-2019. We focus on 2014 and 2015 in this paper because those are the years that we have high-cadence lidar soundings close in time to the peaks in AIRS brightness temperature variance.

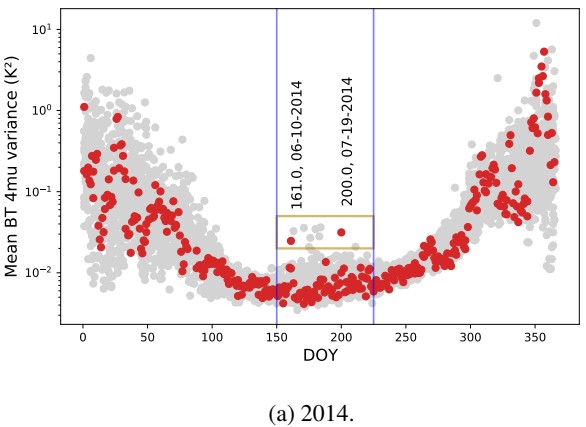

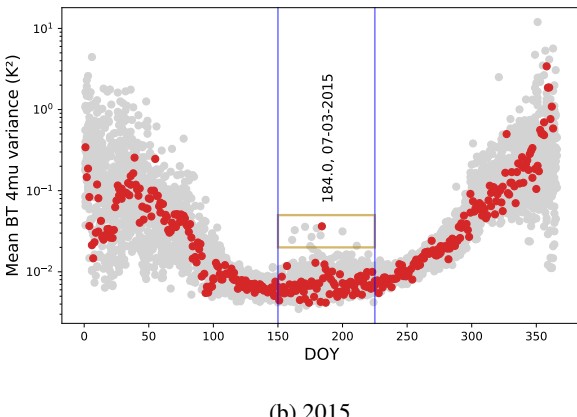

(a) 2014.

(b) 2015.

**Figure 3.** The AIRS mean brightness temperature (BT) variance from the 4 $\mu$m data for the years (a) 2014 and (b) 2015 are shown in red. The mean is taken in a region over Kühlungsborn bounded by 52.35°N and 55.94°N in latitude and 8.68°E and 14.83°E in longitude. In grey, all the data from 2003-2019 is plotted to demonstrate the annual variability of the brightness temperature. The brown rectangle on the right panel shows the peaks in summer we highlight in this study. The blue lines in each panel show the time ranges selected for further study.

To identify deep convection in the region near Kühlungsborn, we apply the method of Aumann et al. (2006) also used by
Hoffmann and Alexander (2010). Aumann et al. (2006) used the 1231 cm$^{-1}$ (8.1 $\mu$m) AIRS radiance channel to identify deep convective clouds between $\pm60°$ of the equator. They argued that because the size of the AIRS footprint at the nadir is 13.5 km, brightness temperatures below 210 K should indicate that the top of the anvil of the thunderstorm protrudes well into the tropopause. The authors point out in the paper that the threshold of 210 K is an arbitrary choice. Deep convective cloud tops typically have horizontal scales of a km or less, and the temperatures of convective cloud tops can be near or below the
temperature of the tropopause, especially when they overshoot the tropopause (Proud and Bachmeier, 2021). This means that even if a footprint contains deep convective clouds, most of the footprint could still be cloud-free. Therefore, the threshold of 210 K, which is well above the temperature of the tropical tropopause, should still identify footprints that contain deep convection. Using the 210 K threshold, they identified about 6000 large thunderstorms per day, almost exclusively between $\pm30°$ of the equator. Hoffmann and Alexander (2010) used a weaker threshold of 220 K to define deep convection over North
America in summer because they found that the threshold of 210 K used by Aumann et al. (2006) was too conservative to identify all deep convective events at mid-latitudes. We increase the threshold even more to 230 K here because the mean tropopause temperature increases to $\sim$220 K in the polar region in summer and Kühlungsborn is on the border between mid- and polar latitudes. For each day, we define deep convection near Kühlungsborn as the fraction of AIRS footprints below the 230 K threshold in a box defined by 0°-12° E and 50°-55° N (shown in the black box in Figure 2; the dark green contour is
230 K). Since convective systems generally move eastward in summer mid-latitudes, we include a larger portion of the defined





region to the west of Kühlungsborn. As mentioned in Section 2.1, we include only AIRS footprints that are in the descending orbit (01:30 local time). There are typically between 1200 and 1600 footprints in the examined area for each day (black box in Figure 2).

To identify gravity wave events in the lidar data, vertical profiles of temperature perturbations are calculated by removing the temperature background. The high- and low-frequency components of the lidar temperature perturbation profile are separated using a fifth-order Butterworth filter with a vertical cut-off frequency of 15 km (Baumgarten et al., 2017) and a temporal cut-off period of 8 hours. This technique ensures that longer-period planetary waves and tidal contributions are removed from the lidar data. After filtering, the GWPED per unit volume is calculated using Equation 1.

$$E_{pV} = \frac{1}{2} \frac{g^2}{N^2} \overline{\left(\frac{T'}{T_0}\right)^2} \overline{\rho} \qquad (1)$$

In Equation 1, $g$, is the gravitational acceleration ($9.8 m/s^2$) and, $\overline{\rho}$, is the daily average atmospheric density profile taken from NRLMSISE-00 (Picone et al., 2002). The Brunt–Väisälä frequency ($N$) is derived from measured background temperature, $T_0$, and $T'$, is the temperature residual. A temporal average is applied over the duration of the measured segment, which is denoted by the over-bar above the temperatures. More details regarding lidar data processing can be found in Baumgarten et al. (2017). To compare the gravity wave activity in AIRS and lidar, we averaged the filtered lidar profiles over 33-43 km

because this is the altitude range where the AIRS 4 $\mu$m kernel function, which has a broad peak.

## 4 Results

In Figure 4, we show the gravity wave activity in AIRS and lidar data for the summers (DOY 150–225) for 2014 and 2015. The dataset from 2003-2019 is shown in the Appendix in Figure A3. The red dots in Figure 4 represent the AIRS mean brightness temperature variance (the summer portion of the time series in Figure 3). Gravity wave activity for the lidar is shown in two

ways: the blue symbols and lines show the vertically filtered (VF) GWPED from the lidar data (vertical wavelength, $\lambda_z < 15$ km), while the black symbols and lines show the temporally filtered (TF) GWPED lidar data (period, $\tau < 8$ hrs). The spatially and temporally filtered lidar GWPED are qualitatively similar (i.e., they show peaks on the same days), however, there are some differences because they represent different parts of the gravity wave spectrum. Gaps in the lidar GWPED time series are due to cloudy weather, which prevented observations. Three major peaks stand out in the AIRS 4 $\mu$m brightness temperature

variance: two peaks in 2014 (DOY 161 and 200) and one peak in 2015 (DOY 184). Major peaks are identified as being above a threshold of 0.02 K$^2$. These peaks are also seen in both the temporally and spatially filtered lidar data around +/- one day. Figure 4 also shows the convective activity in the vicinity of Kühlungsborn (green bars), computed as described in Section 3. The three major peaks in AIRS and lidar gravity wave activity coincide with strong convective activity, which, along with the concentric ring structure of the waves in the AIRS data, suggests these waves are generated by convection.





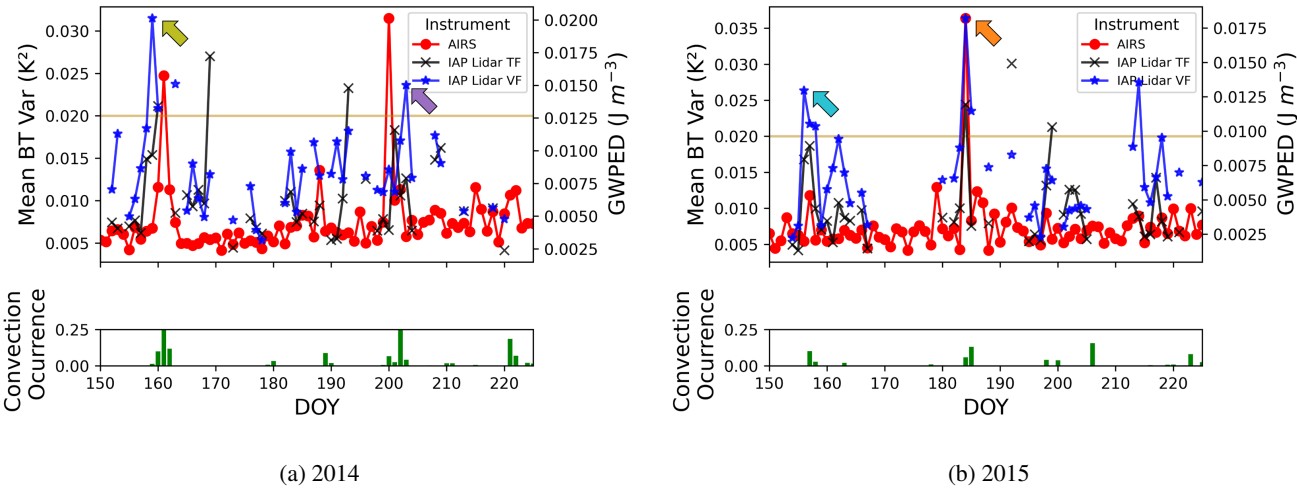

(a) 2014                    (b) 2015

**Figure 4.** Time series (DOY 150-225) of GWPED calculated from lidar observations and brightness temperature variance from the AIRS 4 $\mu$m channel (red circles). The lidar GWPED calculation is done for both temporally (black) and vertically (blue) filtered data. The brown line shows the threshold used to isolate large peaks ($0.02$ K$^2$). The coloured arrows show the profiles that are plotted in Figure 6. In the bottom panel, the convective activity in the area defined in Figure 2 is shown.

There are also several smaller enhancements in Figure 4 where the AIRS brightness temperature variance is still below the $0.02$ K$^2$ threshold. For example, on DOY 157 of 2015 (shown with the cyan arrow), deep convection is recorded, and gravity wave activity is also enhanced in both AIRS and the lidar data. There are also days when deep convection is detected, but there is no corresponding peak in the AIRS brightness temperature variance and unfortunately, there is no lidar data available (e.g., after DOY 220 of 2014 and between DOY 200 and 210 of 2015). Finally, there are several days that the lidar observed increased gravity wave activity and AIRS did not, and in general these are days that do not have deep convection. These events are beyond the scope of this paper because they are related to other gravity wave sources, e.g., baroclinic instabilities.

In Figure 5, we display wave activity, winds, and the deep convection proxy around Kühlungsborn for the largest peak in 2015. The maps show AIRS 4 $\mu$m brightness temperature perturbations in panel 5a, ECMWF winds at 39.85 km altitude in panel 5b, AIRS 8 $\mu$m channel for convection in panel 5c, and ECMWF winds at 10.13 km height in panel 5d. In panel 5a, the characteristic concentric shape of convective gravity waves is observed to the east of Kühlungsborn with brightness temperature perturbations that have amplitudes of around 0.4 K. Gravity waves with longer horizontal wavelengths are mainly seen to the east of Kühlungsborn and gravity waves with shorter horizontal wavelengths are seen to the north of Kühlungsborn. The convective event related to these waves can be seen to the west-southwest of Kühlungsborn with values reaching below 220 K (Panel 5c). Panel 5b shows ECMWF winds at about 40 km altitude approximately 2 hours before the AIRS observation. At the station's latitudes, summer stratospheric winds are generally weak compared to winter, with speeds less than 20 ms$^{-1}$. Although relatively weak, the winds are westward, so that eastward propagating waves are refracted to longer vertical wavelengths and therefore more likely to be observed by AIRS than westward propagating waves (Alexander, 1998).





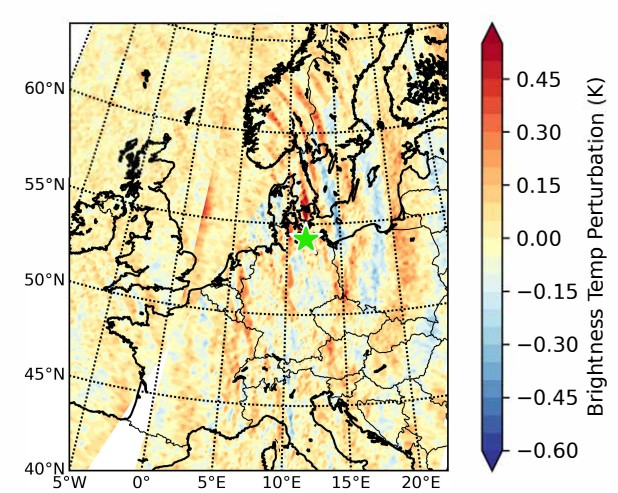

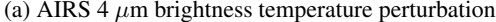

(a) AIRS 4 $\mu$m brightness temperature perturbation

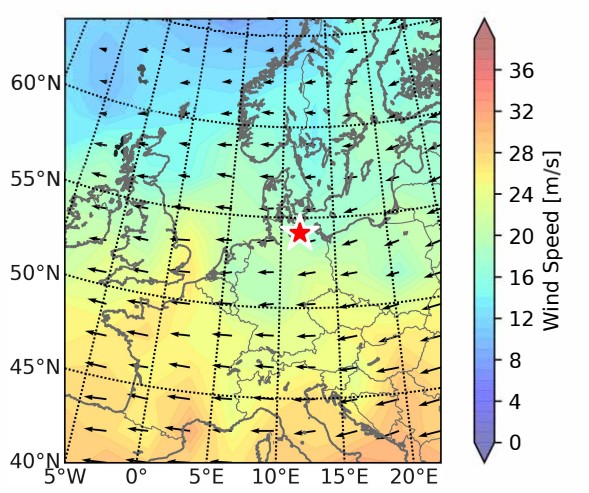

(b) ECMWF wind quivers at around 40 km

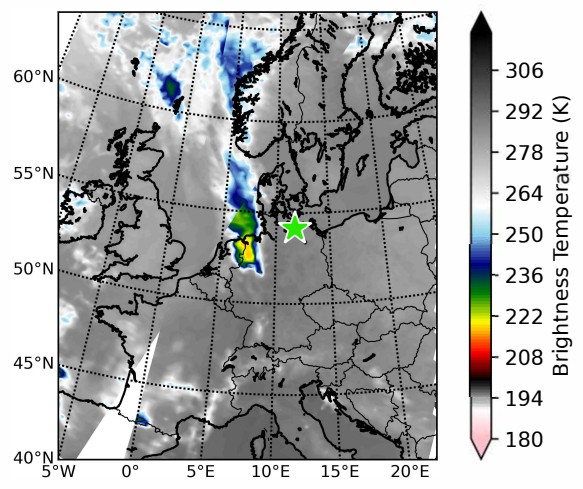

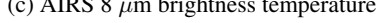

(c) AIRS 8 $\mu$m brightness temperature

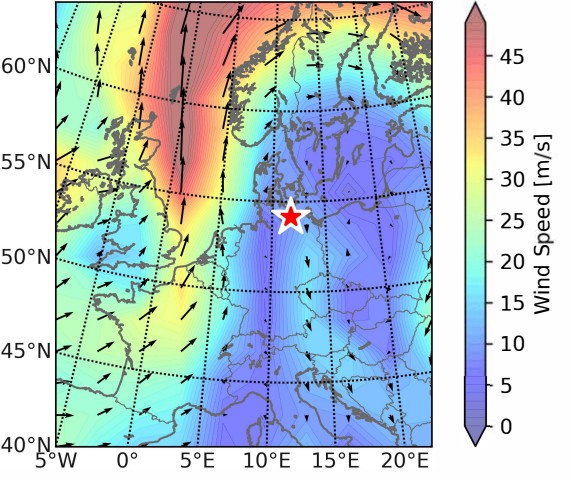

(d) ECMWF wind quivers at around 10 km

**Figure 5.** AIRS brightness temperatures and ECMWF wind data from July 3, 2015 (DOY 184) in horizontal projection are shown. The gravity wave event can be seen moving across the map in an eastward direction in (a), which shows the AIRS 4 $\mu$m brightness temperature perturbation data. A deep convective event associated with the gravity wave event is seen to the west of the Kühlungsborn (marked with a green star) in (c). (b) shows ECMWF winds at 39.85 km altitude. A jet north-west of Kühlungsborn is shown in ECMWF winds at 10.13 km (d).





The half-ring features in the 4 $\mu$m brightness temperature can be explained by two possible mechanisms. Using a simulation, Alexander (1996) showed a strong preference for westward propagating waves with a westward propagating convection cell in the troposphere. In the cases presented here, the convective systems are moving eastward and AIRS observes eastward propagating gravity waves (propagating against the mean winds). The AIRS observational filter also impacts which gravity waves are seen in the 4$\mu$m brightness temperatures. AIRS is more sensitive to gravity waves with longer vertical wavelengths. Since the background wind near 40 km in the summer is westward, eastward propagating gravity waves are refracted to longer vertical wavelengths and are observed by AIRS.

In Figure 5d, which shows the winds at around 10 km altitude, there is an upper tropospheric jet extending from the North Sea to the west coast of Norway and into the Arctic (part of a larger ridge pattern). There is a frontal system at the trailing edge of the eastward propagating ridge, which is where the convection is strongest in Figure 5c. In the areas where the wind speed is 20 m/s or weaker around 10 km altitude (area around 7°E-23°E and below 60°N), AIRS observes gravity waves. A similar scenario is observed in all three cases (the other two cases can be found in Appendix Figure A1 and Appendix Figure A2). The strong winds of the jet could be affecting the westward propagating portion of the gravity waves generated by the convection. Or, as stated above, they are simply not observed by AIRS because of the westward winds in the stratosphere.

In Figure 6, we show the vertical profiles of GWPED derived from the lidar temperatures for the summers of 2014 (Figure 6a and 6b) and 2015 (Figure 6c and 6d) and highlight the vertical profiles of the events discussed above. These highlighted profiles correspond to the highest peaks of Figure 4 correlated with convective activity. Temporally filtered data is shown in the left panels (a and c) and vertically filtered data in the right panels (b and d). The grey profiles show all the data from each specific year. The red highlighted area in all plots shows the profile heights that were averaged and plotted in Figure 4 (33-43 km). The olive colour shows the maximum peak in the vertically filtered lidar data (DOY 159) corresponding to the first peak of 2014 in AIRS, and the purple colour shows the second maximum in the vertically filtered lidar data (DOY 203). Note that the peaks in 2014 did not occur on the exact same days for AIRS and the lidar. For the first peak, this is because the lidar did not measure on the day of the AIRS peak. For the second peak, it could be because of the different filters applied to the data (e.g., the peaks in the temporally and vertically filtered lidar data also don't occur on the same days). The orange colour highlights the vertically filtered lidar profile with the largest GWPED in Figure 4b for 2015. We highlight an additional convective event in 2015 that is under the threshold of 0.02 K$^2$ in AIRS brightness temperature variance, but still has a local maximum in both AIRS and lidar (DOY 157). Even though it doesn't exceed our AIRS threshold, it stands out as an event associated with convection that is seen in all three-time series. The vertically filtered data shows an enhancement in GWPED in the region of study in all the cases. These are among the largest values in the designated height range.

## 5 Discussion

Given the intermittent observations of the lidar and the different observational and processing filters (Figure 1), it's remarkable that the peaks in the AIRS 4$\mu$m brightness temperature coincide with the wave activity seen by lidar (Figure 4). Hoffmann and Alexander (2009) showed that AIRS 4$\mu$m brightness temperatures are most sensitive to vertical wavelengths longer than 15





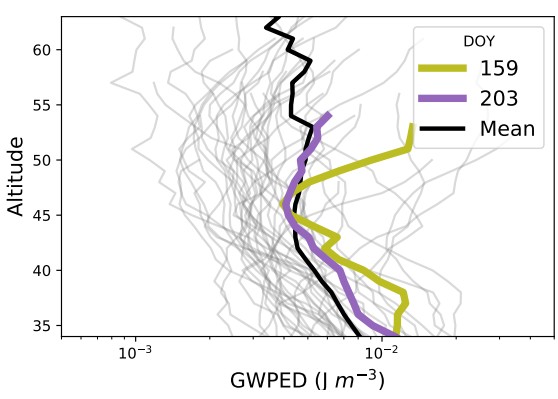

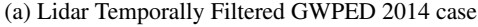

(a) Lidar Temporally Filtered GWPED 2014 case

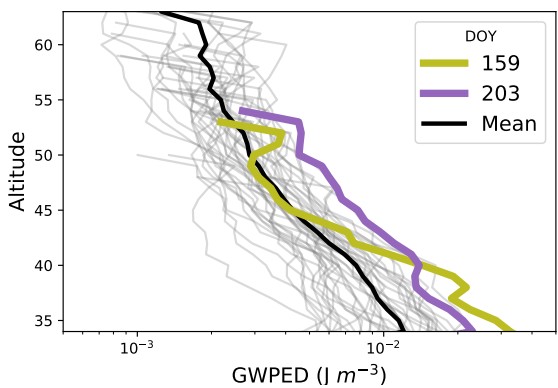

(b) Lidar Vertically Filtered GWPED 2014 case

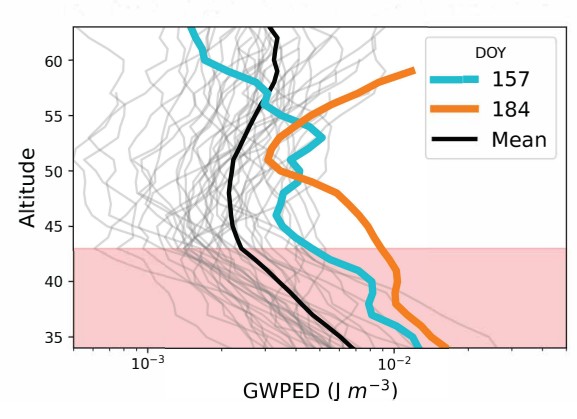

(c) Lidar Temporally Filtered GWPED 2015 case

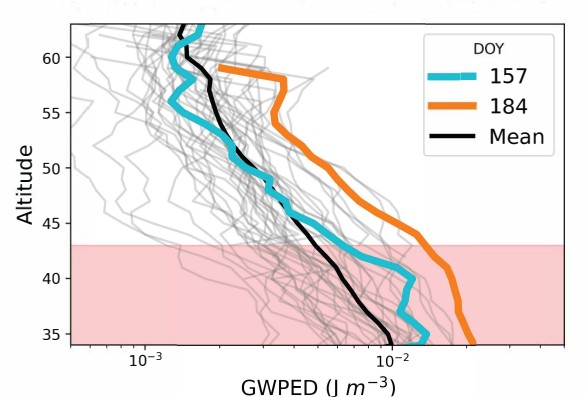

(d) Lidar Vertically Filtered GWPED 2015 case

**Figure 6.** Vertical profiles of GWPED calculated using the temporally (left panel) and vertically (right panel) filtered lidar data are shown. The top panel shows the results for the case year 2014. The bottom panel shows the results for the case year 2015. In all panels, the grey lines in the background of each figure show the rest of the days that we have data for the summer period (DOY 150-225), and the thick black line shows the average of all the profiles. The red highlighted area shows the heights that were averaged to get the results seen in Figure 4 (33-43 km)



km. So the agreement is all the more surprising since each system observes a different part of the gravity wave spectrum. It's counter-intuitive that the vertically filtered lidar data agrees better with AIRS, given that they should have very little overlap in vertical wavelengths. This suggests that a broad range of vertical wavelengths is present in the convective gravity waves. Several studies have shown that convection produces gravity waves with dominant vertical wavelengths of approximately

twice the depth of the latent heating associated with the convection (e.g., Vadas and Fritts, 2001; Pandya and Alexander, 1999). However, other studies have shown that although the maximum gravity wave response generally has a vertical wavelength of approximately twice the depth of the heating, there can still be substantial power at longer vertical wavelengths (Holton et al., 2002). Our results also support this conclusion.

The vertical profiles of GWPED show an enhancement above the mean profiles in the region of study in all the peak cases,

especially the vertically filtered profiles. Except for the temporally filtered profiles in 2014 (Figure 6a), the profile values are among the largest in the designated height range. The general shape of the profiles with altitude has been discussed already by Strelnikova et al. (2021). In the absence of gravity wave sources in the middle atmosphere, we expect that GWPED should decrease with height above the tropopause as gravity waves generated in the troposphere dissipate. The increase in GWPED above 45 km in the temporally filtered lidar is interesting and could imply secondary gravity wave generation; however, it

is difficult to speculate on the shape of the vertical profiles of GWPED because they could be influenced by the spatial and temporal filters used.

The events shown in this work occur in the AIRS data 9 times in 17 summers. These events are collected and shown in Table 1. This number of events is only for the area of our study around Külungsborn, although more events might occur that were not detected with the area we selected. The amplitudes of the AIRS 4 $\mu$m brightness temperature anomalies are not remarkable

when compared to those that AIRS observes in winter and especially over orography. However, we know that AIRS is much more sensitive to gravity waves with longer vertical wavelengths, which means that AIRS observes stronger gravity wave activity when the background winds are large (i.e., winter). If we look at the values of the peaks of GWPED in the vertically filtered lidar data in Figure 4, they range between ∼0.015-0.02 J m$^{-3}$. These values exceed the mean summer values in the climatology presented in Strelnikova et al. (2021) with the same lidar instrument. They also approach the values observed in

the winter months (JFM), where there is a seasonal maximum. We know from observations and high-resolution modelling studies of gravity waves that GWPED and gravity wave momentum flux have an approximately log-normal distribution, such that events in the tail of the distribution occur more rarely but contribute significantly to the total zonal mean forcing from gravity waves (e.g., Hertzog et al., 2008; Jewtoukoff et al., 2015; Holt et al., 2017; Strelnikova et al., 2021). These events are potentially in the tail of the momentum flux distribution and could contribute significantly to the zonal mean forcing in the

stratosphere in summer. We will investigate this in more detail in future studies.

In this work, we show that large gravity wave events detected by both AIRS and lidar coincide with convective activity. Another clue that these events are convectively generated is the half-ring shapes observed in the AIRS 4 $\mu$m brightness temperatures. Further, supporting the case for convective sources is that the topography is mostly flat in our region, which rules out orographically generated gravity waves. Jet imbalance is an important gravity wave source, as discussed by Hoffmann

and Alexander (2010). However, waves identified in the stratosphere emanating from regions of jet imbalance have shorter



**Table 1.** Days in 2003-2019 where the mean brightness temperature variance from AIRS $4\mu$m channel over Kühlungsborn region exceeds $0.02\ \mathrm{K}^2$ (values inside the red rectangle of Figure 3b). The days in this table are based on Figure A3 in the Appendix.

| Year | DOY |
|------|---------|
| 2005 | 211 |
| 2011 | 180 |
| 2012 | 183 |
| 2013 | 171 |
| 2014 | 161, 200 |
| 2015 | 184 |
| 2016 | 176 |
| 2019 | 163 |

vertical wavelengths than can be observed by AIRS (e.g., Plougonven and Snyder, 2007). A relatively strong jet associated with frontal systems that generated deep convection was present in all the cases in our study. Above where the jet is observed, gravity waves are not observed in the AIRS $4\mu$m channel. The gravity wave spectrum is especially sensitive to wind shear in the upper troposphere (Beres et al., 2002), however further studies are needed to understand how the jet is influencing the
observed events.

## 6 Conclusions

In this paper, we identified a general increase in gravity wave activity during the summer above Kühlungsborn, Germany, which we connect to convective events over Western Europe. As seen in Figure 3, this increase of around one order of magnitude observed in brightness temperature variance of the AIRS 4 $\mu$m emission is composed of sporadic gravity wave events that
occur in about half of the years since the launch of the AQUA satellite in 2002 (see Table 1 and Figure A3). We chose two years with events that had such peaks in brightness temperature variances in AIRS to investigate in more detail based on the availability of lidar data. We showed that these events were also detected in lidar data (both vertically and temporally filtered). Additionally, we showed that for the summer period, the gravity waves observed in AIRS and the lidar are likely generated by convection. We used AIRS $8\mu$m brightness temperature to identify deep convection and showed that the large gravity wave
events detected by both AIRS and lidar coincide with convective activity.

We also showed profiles of GWPED for days of peak gravity wave activity in the lidar. We showed that there is an enhancement in the altitude range of 33-43 km for all days with convective events when compared to the mean in the vertically filtered data. These values exceed the mean summer values in the climatology. They also approach the values observed in the winter months (JFM), where there is a seasonal maximum. These events, while infrequent, could provide significant forcing
in the stratosphere and are not currently accounted for in weather and climate models. Further studies of convective gravity wave events are required to improve their representation in models, which would in turn improve forecast skill and accuracy

from weather to climate. Recently, we have developed a Rayleigh Doppler wind lidar in the same location as the lidar used in this work. This allows us to measure in-situ winds and be able to expand our work on gravity wave activity near Küh-lungsborn, Germany. This new system and dataset will bring us closer to understanding the summer gravity wave activity over
Kühlungsborn.

*Data availability.* RMR lidar data used in this publication can be found at the following data archive

https://www.radar-service.eu/radar/en/dataset/MrBhYNcasORKhVEx?token=UtHPWXA. ECMWF data is freely available at

https://www.ecmwf.int/en/forecasts/access-forecasts/access-archive-datasets. AIRS plots are freely avail-

able courtesy of Dr. Lars Hoffmann https://data.fz-juelich.de/dataset.xhtml?persistentId=doi:10.26165/JUELICH-DATA/LQAAJA

*Author contributions.* E.F.D wrote the software to conduct the majority of the formal analysis, made all of the figures, and composed the
original draft. M.G. provided the lidar data, supervision, interpretation of the results, scientific insight, and extensive suggestions to improve the original manuscript. L.H. contributed to the analysis of the AIRS data, interpretation of the results, scientific insight, and extensive sug-gestions to improve the original manuscript. I.S. provided scientific insight, interpretation of the results, and extensive suggestions to improve the original manuscript. R.W. provided scientific insight and extensive suggestions to improve the original manuscript. G.B. contributed to the processing of ECMWF data, supervision, and scientific insight. F-J.L. provided supervision, scientific insight, and suggestions to strengthen
the abstract and conclusions.

*Competing interests.* Franz-Josef Lübken is a member of the editorial board of ACP.

*Acknowledgements.* This paper is a contribution to the project Analyzing the Motion of the Middle Atmosphere Using Nighttime RMR-Lidar Observations at the Midlatitude Station Kühlungsborn (AMUN) funded by Deutsche Forschungsgemeinschaft (DFG) - Projektnummer 445400792. L.H. was supported by NASA Grant 80NSSC20K0950. We thank Lars Hoffmann for the AIRS stratospheric gravity wave dataset
(Hoffmann, 2021).

# Appendix A: Other Cases





(a) AIRS 4 $\mu$m perturbation

(b) ECMWF wind quivers near 40 km

(c) AIRS 8 $\mu$m convection

(d) ECMWF wind quivers near 10 km

**Figure A1.** AIRS brightness temperatures and ECMWF wind data from June 10, 2014 (DOY 161) in horizontal projection is shown. The gravity wave event can be seen moving across the map in a north-eastward direction in Panel (a), which shows the AIRS 4$\mu$m Perturbation data. A deep convective event associated with the gravity wave event is seen over the Netherlands, specifically to the southwest of the Kühlungsborn (marked with a green star) in Panel (c). (b) shows ECMWF winds of around 20 m/s at 39.85 km altitude. A jet close to the region of Kühlungsborn is shown in ECMWF winds at 10.13 km(d).



(a) AIRS 4 μm Perturbation

(b) ECMWF wind quivers at around 40 km

(c) AIRS 8 μm Convection

(d) ECMWF wind quivers at around 10 km

**Figure A2.** AIRS brightness temperatures and ECMWF wind data from July 19, 2014 (DOY 200) in horizontal projection are shown. The gravity wave event can be seen moving across the map in a north-eastward direction in Panel (a), which shows the AIRS 4μm Perturbation data. A deep convective event associated with the gravity wave event is seen over the western part of France, specifically to the southwest of the Kühlungsborn (marked with a green star) in Panel (c). (b) shows ECMWF winds of around 20 m/s at 39.85 km altitude. A jet close to the region of Kühlungsborn is shown in ECMWF winds at 10.13 km (d).





**Figure A3.** AIRS 8 $\mu$m deep convection (green bars) and 4 $\mu$m brightness temperature variances (red circles) averaged over the regions confined by the black and green boxes in Figure 2, respectively. The convection corresponds to the left axis and the brightness temperature variance corresponds to the right axis. The brown line shows the 0.02 K$^2$ threshold chosen to highlight extreme gravity wave events.



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
