# Peer review of "Convective gravity wave events during summer near 54°N, present in both AIRS and RMR Lidar observations"

_EGUsphere, 2023_

## Author Comment (AC1)

Thank you for taking the time to help improve our manuscript. We have addressed all requested points in **red**.

The paper by Franco-Diaz et al. is an interesting study that combines lidar observations of gravity waves at Kuhlungsborn with NASA's AIRS satellite observations of deep convective clouds in one infrared channel and observations of gravity waves in another infrared channel. The focus of the paper are convective events during summer upstream of Kuhlungsborn that excite gravity waves propagating downstream over Kuhlungsborn. One main finding is that for several strong convective events convective gravity waves are seen at the same time by both the lidar and AIRS although the observational filters of the two instruments are very different. This supports the assumption that convective sources emit a broader spectrum of gravity waves. The authors suggest that sporadic strong events of convective gravity waves should still be important for middle atmosphere dynamics because midlatitude gravity wave activity during summer is quite weak and the events are much stronger than the monthly average gravity wave activity.

Overall, the paper is well written, the figures are adequate, and the topic is of great interest for the readership of ACP. The paper is therefore recommended for publication in ACP after addressing my minor comments.

Main comments are:

(1) There is some confusion about the AIRS observational filter. Please check the paper for consistency and refer to existing literature.

- We addressed this issue at different places of the manuscript. AIRS is most sensitive to waves of more than 26 km vertical wavelength, as described in the data set linked to Hoffmann (2021). Thank you for pointing this out.

(2) Some readers may be confused by introducing an area for identifying deep convective clouds far upstream of Kuhlungsborn. Therefore some more reasoning should be given earlier in the paper why this selection is made.

 - We added some justification in Section 3 (line 187) where the black box is first introduced. The new text is as follows:

 "Since convective systems generally move eastward in summer mid-latitudes and gravity waves propagate radially from the convective center, we chose the region defined by the black rectangle to include a large area to the west of Kühlungsborn."

Specific Comments:

(1) l.5 Here you just write "using ECMWF". Which product? analyses, forecasts, reanalyses?

– We us the ECMWF Integrated Forecast System (IFS) operational analysis. This is now addressed on line 5 of the revised paper.

(2) l.36: The reference Marlton et al. (2021) is about the effect of assimilating observations in general - not about convective gravity waves. You should at least add one or two papers showing the importance of convective gravity waves, for example Kim et al. (2013) and Bushell et al. (2015).

Kim, Y.-H., A. C. Bushell, D. R. Jackson, and H.-Y. Chun (2013), Impacts of introducing a convective gravity-wave parameterization upon the QBO in the Met Office Unified Model, Geophys. Res. Lett., 40, 1873-1877, doi:10.1002/grl.50353.

Bushell, A. C., Butchart, N., Derbyshire, S. H., Jackson, D. R., Shutts, G. J., Vosper, S. B., and Webster, S.: Parameterized gravity wave momentum fluxes from sources related to convection and large-scale precipitation processes in a global atmosphere model, J. Atmos. Sci., 72, 4349-4371, 2015.

– We thank the reviewer for this comment. This mention the suggested references on line 36 of the revised paper.

(3) l.64: Here you should add more references to previous work about arc-shaped gravity wave patterns. For example, Gong et al. (2015) performed a global survey of concentric gravity waves seen by AIRS, showing that such patterns occur at midlatitudes. Another example is Ern et al. (2022). In this study an arc-shaped gravity wave pattern is related via backward raytracing to deep convection and latent heat release caused by the 2022 Tonga volcanic eruption.

Gong, J., J. Yue, and D. L. Wu (2015), Global survey of concentric gravity waves in AIRS images and ECMWF analysis, J. Geophys. Res. Atmos., 120, 2210-2228, doi:10.1002/2014JD022527.

Ern, M., Hoffmann, L., Rhode, S., & Preusse, P. (2022). The mesoscale gravity wave response to the 2022 Tonga volcanic eruption: AIRS and MLS satellite observations and source backtracing. Geophysical Research Letters, 49, e2022GL098626. https://doi.org/10.1029/2022GL098626.

– Thank you for the suggestions and references. We added them on line 65 of the revised paper.

(4) l.96: The sensitivity of the AIRS 4mu channels is rather Lz>30km than Lz>15km, see Fig.3d in Hoffmann and Alexander (2009). Further, the number of 15km does not match with the 26km given in your Fig.1. Please check for consistency!

– We apologize for the confusion. AIRS is sensitive to waves with Lz > 26 km as described by Hoffmann (2021), We changed it throughout the whole paper.

(5) l.100: You should point out the importance of using observed deep convective clouds.

As has been shown by Aumann et al. (2023) deep convective clouds in meteorological forecasts, for example the ECMWF IFS, are less reliable.

Aumann, H. H., Wilson, R. C., Geer, A., Huang, X., Chen, X., DeSouza-Machado, S., and Liu, X.: Global Evaluation of the Fidelity of Clouds in the ECMWF Integrated Forecast System, Earth and Space Science, doi:10.1029/2022EA002652, 2023.

 – Thanks again for the recommended citation and suggestion. It has been incorporated on line 105 of the revised manuscript.

"In this work, we use the observed deep convective clouds instead of the forecast. [Aumann2023] has shown that the ECMWF IFS deep convective clouds are less reliable than the observed."

(6) l.131/132: "corresponds to ~500km" - not clear what this means! Please be more specific!

By subtracting a 4th-order across-track polynomial, horizontal wavelengths in the approximate range 30km to 1000km should be still in the AIRS data. An approximate sensitivity function is given, for example, in Meyer et al. (2018), Fig. 3a.

This sensitivity function applies to retrieved temperatures. Therefore the magnitude of the sensitivity for brightness temperatures will be quite different, but relative variations in the horizontal wavelength direction should be similar because a 4th order polynomial was applied in both cases. At short horizontal wavelengths the sensitivity is limited by the size of the AIRS footprints.

Meyer, C. I., Ern, M., Hoffmann, L., Trinh, Q. T., and Alexander, M. J.: Intercomparison of AIRS and HIRDLS stratospheric gravity wave observations, Atmos. Meas. Tech., 11, 215-232, https://doi.org/10.5194/amt-11-215-2018, 2018.

- Thank you for pointing this out. We removed the part of the sentence you referenced, and since we already described the horizontal wavelength sensitivity of the AIRS brightness temperatures in the AIRS section, we decided not to repeat it here.

(7) Fig.1: Where do the approximate sensitivity ranges for AIRS come from? Are they adapted from Fig.3d in Hoffmann and Alexander (2009)?

- Yes, the sensitivity ranges are obtained from this reference and also the response functions from the data repository: Hoffmann, Lars, 2021, "AIRS/Aqua Observations of Gravity Waves", https://doi.org/10.26165/JUELICH-DATA/LQAAJA, Jülich DATA, V1. We have edited the text to make things more clear.

"This sketch shows which vertical wavelengths the AIRS brightness temperatures and the temporally and vertically filtered lidar data are sensitive to. The response function for AIRS was obtained from the midlatitude function provided by [Hoffmann2021]. Note that the response function for AIRS for vertical wavelengths below ~26 km is very small (< 1\%). While this idealized figure implies hard cut-offs at certain vertical wavelengths, the nature of

the filters (whether observational or imposed) is more of a gradual transition."

(8) l.141: Again, the 15km apply rather to the AIRS 15mu channels than to the 4mu channels (see Fig.3 in Hoffmann and Alexander (2009)). Further, the number of 15km is inconsistent with your Fig.1

- In Figure 1, the sensitivity to vertical wavelengths less than about 26 km is very small (we referenced in the text box that it is less than 1%, and the very light color shading is meant to indicate this low sensitivity. We have pointed this out in the new figure caption:

"… Note that the response function for AIRS for vertical wavelengths below ~26 km is very small (< 1\%). While this idealized figure implies hard cut-offs at certain vertical wavelengths, the nature of the filters (whether observational or imposed) is more of a gradual transition."

(9) l.171: Another reason why selecting a region west of Kuhlungsborn for detecting deep convective clouds makes sense is because gravity waves will propagate radially away from the convective center. However, only the gravity wave structures that propagate opposite to the prevailing stratospheric westward wind (during summer) will become visible for AIRS because their vertical wavelengths are refracted towards larger values by Doppler-shifting.

- This is a good point and we have added your suggestion to the text. The discussion about the portion of the wave that is propagating against the background wind being visible by AIRS is discussed in Section 4.

"Again, since convective systems generally move eastward in summer mid-latitudes and gravity waves propagate radially from the convective center, we include a larger portion of the defined region to the west of Kühlungsborn."

(10) l.176: Your sentence reads as if two filters are applied simultaneously, which I think is not the case. Suggestion:

using a fifth-order Butterworth filter with a vertical cut-off frequency of 15 km (Baumgarten et al., 2017) and a temporal cut-off

->

using either a fifth-order Butterworth filter with a vertical cut-off frequency of 15 km (Baumgarten et al., 2017), or a temporal cut-off

- Thank you for the suggestion, we have made the change.

"The high- and low-frequency components of the lidar temperature perturbation profile are separated using either a fifth-order Butterworth filter with a vertical cut-off frequency of 15 km [Baumgarten2017], or a temporal cut-off period of 8 hours.

(11) l.196: As can be seen from Fig.4, major peaks in the lidar data can be separated by up to three days (not by just one day as stated in l.196). Please comment!

- We corrected the text to say that the offset can be up to three days. A discussion of this is provided at the end of Section 4 already, so we did not add further comment on line 196.

"These peaks are also seen in both the temporally and spatially filtered lidar data, but the peaks in the lidar data are offset by up to 3 days."

(12) l.197: Here you write "vicinity of Kuhlungsborn", which is a bit misleading. Better refer to the area given by the black rectangle in Fig.2!

- We have made the suggested change.

(13) l.206: Could it also be that convection outside the black rectangle could have caused the gravity waves?
Please note that even for the wave event on 03 July 2015 Kuhlungsborn is just at the edge of the wave pattern seen by AIRS (see Fig.5).

- Yes this is a valid point. We have added this to the text as a possibility.

"These events are beyond the scope of this paper because they are either generated by convective activity outside of the area we chose for this study or by other gravity wave sources, e.g., baroclinic instabilities."

(14) l.207: Again, "around Kuhlungsborn" may be misleading! Please refer to the black rectangle!

- Thank you for the suggestion. In Figure 5, we are showing the variables for a large region centered over Kühlungsborn. We have changed "around Kühlungsborn" to "centered over central Europe" in an effort to make it clearer.

(15) l.250/251: longer than ~30km for the 4mu channels

- We have changed this everywhere to ~26 km since this corresponds to where the AIRS sensitivity starts to increase in our Figure 1. Figure 1 was produced using the response function of the AIRS 4 mu channel from Lars's website.

(16) l.255: Please cite also the earlier work by Salby and Garcia (1987) who introduced the depth of the heating concept:

Salby, M. L. and Garcia, R. R.: Transient response to localized episodic heating in the tropics, Part I: Excitation and short-time near-field behavior, J. Atmos. Sci., 44, 458-498, 1987.

- Thank you, we have added the reference. (line 311)

(17) l.278: A recent climatology of gravity wave intermittency is given in Ern et al. (2022). In this paper it is shown that the gravity wave distribution in the summer hemisphere is even more intermittent than in the tropics, which supports your findings of sporadically occurring strong convective gravity waves at midlatitudes during summer.

Ern, M., Preusse, P., and Riese, M.: Intermittency of gravity wave potential energies and absolute momentum fluxes derived from infrared limb sounding satellite observations, Atmos. Chem. Phys., 22, 15093-15133, https://doi.org/10.5194/acp-22-15093-2022, 2022.

- We have added the reference to the manuscript along with some additional text.

"[Ern2022] further show that the gravity wave distribution in the summer hemisphere is even more intermittent than in the tropics. This supports the idea that the events presented here are in the tail of the momentum flux distribution and could contribute significantly to the zonal mean forcing in the stratosphere in summer."

(18) l.284: Another point worthwhile mentioning is that stratospheric winds are relatively weak when the wave events are detected. Orographic gravity waves should therefore have relatively short vertical wavelengths and should therefore be invisible for AIRS. The same should hold for the jet-generated gravity waves as you mention later. For orographic gravity waves there may even exist critical wind layers at midlatitudes during summer.

- The suggestion has been incorporated to the manuscript in the paragraph beginning on what is now line 330.

Technical Comments:

l.111: systems -> system

- Done

Fig.1: sensitivity for AIRS should read Lz ">" ~26km  (not "<" 26km)

- Thank you for noticing this mistake. The caption is rewritten according to a previous comment.

l.158: tropopause -> tropopause region ???

- The change was made.

l.185: ???

kernel function, which has a broad peak.

->

kernel function has a broad peak.

- Sorry for this sentence. We changed it to "To compare the gravity wave activity in AIRS and lidar, we averaged the filtered lidar profiles over 33-43 km. The reason behind this selection was because this altitude range is where the broad AIRS 4 µm kernel function peaks.

Fig.4, lower panels: Ocurrence -> Occurrence

- We have made the suggested correction.

l.243: under the threshold -> below the threshold

- We have made the suggested change.

Fig.A3, y-axis: Ocurrence -> Occurrence

- We have made the suggested correction.

l.349: filtermethods -> filter methods

- Done

l.406/407: Title of this paper is not correct, please check!

- Thanks for finding this error. We correct it in the revised version.

---

## Author Comment (AC2)

Thank you for taking the time to help improve our manuscript. We have addressed all requested points in **red**.

This manuscript by Franco-Diaz investigates convective gravity wave effects over northern Europe using data from NASA's AIRS satellite instrument, a Rayleigh-Mie radar at Kuhlungsborn, and supporting data from ECMWF operational data.

The manuscript is well-written, clear and quite an easy read, but makes interesting and well-evidenced points, showing some nice case studies. I therefore echo Reviewer 1 in their recommendation that only minor corrections are needed for publication in ACP. In particular, the paper was well-structured, the figures well-chosen, and in general most questions I thought of while reading were answered within a couple of sentences at most.

I include a series of minor comments below. I also strongly echo the comments of Reviewer 1, particularly their comments:

(1) about consistency in the description of AIRS' observational capabilities - for example, lines 95, 131, and 140 disagree on the horizontal-wavelength sensitivity of the products used, while (as reviewer 1 says) line 96 disagrees with other parts about vertical sensitivity.

and (10) about filters, as I also read it the same way

– We have modified references to the resolution throughout the manuscript to be consistent with Figure 1

In addition, I would recommend/ask that the authors:

(A) be much more specific about the ECMWF data used, particularly in the abstract where the description given ("using ECMWF") is really quite undescriptive! Reviewer 1 asks for more details of the dataset; I would also like the authors to check that they really mean ECMWF *forecast* data as they say on line 119 - do they mean operational *analysis* data? Using the operational analysis, i.e. ECMWF's best-guess of the atmospheric state, would make perfect sense, while forecast data, which is generated by running the model forward from the analysis, would be a somewhat esoteric option to use for a study like this.

- Thank you for the comment. We use the ECMWF IFS observational analysis. It has been clarified in the paper.

(B) in general, the paper does not reference many other studies which have looked at the same gravity waves in multiple datasets. Discussing a few of these in the introduction could be useful to contextualise the observational filter differences you see - I would naturally recommend our 2016 paper on orographic waves (https://amt.copernicus.org/articles/9/877/2016/amt-9-877-2016-discussion.html), but studies by many others are of course available on the same topic.

- Thank you for the suggestion. We have added a paragraph to the introduction and incorporated the reference you gave.

"Simultaneous observations from different instruments of the same gravity wave event are useful for providing insight into different portions of the gravity wave spectrum since no single instrument is capable of viewing the entire gravity wave spectrum. Each measurement technique has its strengths and limitations. Lidars have very high temporal and vertical resolution but only measure at one location. Limb sounders have good vertical but poor horizontal and temporal resolutions. Nadir viewing satellite instruments have good horizontal but poor vertical and temporal resolutions. Observations of gravity wave properties from various instrument types can differ considerably because each measurement technique is sensitive to different parts of the wave spectrum (observational filter). For example, [Wu2006] found that most of the differences in gravity wave variance distributions between different types of instruments could be related to their viewing geometry and thus their different sensitivities to various portions of the gravity wave spectrum. Similarly, [Wright2016] found that gravity wave properties for the same event over the Drake Passage measured by the nadir-viewing AIRS instrument, radiosondes, radar, and limb sounders differed significantly, sometimes being entirely uncorrelated, suggesting that the discrepancies were due to the different observational filter of each instrument. Typically, there is good agreement between instruments of the same type or that measure similar parts of the gravity wave spectrum [Wright2016, Ern2018]. Good agreement has also been shown when sampling one instrument to match the resolution of another. For example, [Preusse2000] showed that CRISTA gravity wave zonal mean variance was comparable to that of MLS if CRISTA vertical resolution was reduced to MLS vertical resolution. Understanding the full spectrum of gravity waves generated by convection requires combined analysis of instruments measuring different parts of the gravity wave spectrum and, as mentioned above, high-resolution simulations. In this study, we combine gravity wave observations over the same geographical location from two very different types of instrument: lidar and nadir-viewing AIRS. We focus on case studies of strong convective gravity wave activity observed by both instruments in the summers of 2014 and 2015."

(C) does the 8-hour limit cutoff a chunk of permitted GW periods at this latitude? Presumably, going by the text, the 8h cutoff is chosen to avoid the mesospheric terdiurnal tide impacting the results, but in the 33-43km part of the stratosphere this isn't a major concern as it's so weak in temperature amplitude this far down in the atmosphere

- Yes, we are aware of this potential shortcoming. First, we used this filter for consistency with our previous publications of the lidar data [Baumgarten et al., JGR, 2017, and Strelnikova et al., JAS, 2021]. Indeed, the temporally filtered data will probably cut part of the inertia gravity waves. That is why we also show the vertically filtered data where these waves are included. Also, the influence of the tides in lidar data should be very small in the lower stratosphere (less than 0.5 K) [Hauchecorne et al. 2019]. We used a standard lidar data product optimized for retrievals in the mesosphere where tidal contributions are more relevant. In future work we will modify the stratospheric temperature retrieval to remove the unnecessary filter. We have added the following caveat to our manuscript on line 219:

"It is important to note that some longer period GWs will also be affected by the use of the filter."

Some additional minor comments follow, but in general I think this is a very good paper and in my view is publishable in something very close to its current form. Good work!

Additional comments:

014 dimensions unclear - should specify these are horizontal wavelengths

- We have made the correction.

018 and also unresolved

- We have made the correction.

077 and afterwards - Aqua is a name, not an acronym, so shouldn't be all-caps

- We have made the correction.

079: would "at least two" work better here?

- We have made the correction.

085: the track-edge value would be useful as it is quite a bit lower (~40km)

- We have made the addition. The text now reads:

"This scan width is composed of 90 footprints that have a diameter of 13.5 km at nadir and increase in size off-nadir (~40 km at the edge of the scan)."

089: how is the product "special"? This is quite a vague way of describing the dataset.

- We have deleted the word "special".

093: I *think* (but a happy to be corrected) that the 4poly method needs referencing to Alexander + Barnet 2003 (JAS)

- Thank you for this comment. We have added the reference and slightly expanded the description. It reads now "The brightness temperature anomalies in this product were obtained by fitting and subtracting a fourth-order polynomial to the cross-track radiances to remove the large-scale background as well as limb brightening effects [Alexander2006].

098: brackets missing from ref

- We have made the correction.

101: what do you mean by "atmosphere" here? Couldn't one say that the clouds you're measuring are part of the atmosphere, just a wetter part?

- Sorry, we do not understand this comment.

108: Can MST radar do this?

- MST radars do measure winds in the troposphere and lower stratosphere, but no temperatures. At certain conditions (i.e. the existence of PMSE) they measure winds also in the mesopause region. There exist other radar techniques for the MLT region, but they are typically also confined to wind measurements. Lidar is the only technique that covers the full range mentioned here and is capable of temperature sounding. While it might be possible to measure winds in the stratosphere with radar, it is very difficult in practice and requires long integration times and large arrays [Maekawa et al., 1993]. The radar detects backscatter signals from strong refractive index fluctuations in the atmosphere. The refractive index of the atmosphere is a function of three components: dry air (air density), water vapor (humidity) and free electrons [Balsley and Gage 1980; see also Fig. 1 of Kato 2009]. The reason why it is very hard for radars to observe the mid- to upper stratosphere is because the radio refractive index has a minimum at altitudes of around 20-60 km as air density is decreasing and electron density is still low. [Woodman1974] shows the radar gap in the profile of echo power versus height obtained by the Jicamarca Radar in their Figure 2. We are only aware of one existing publication of radar measurements of vertical winds in the stratosphere [Maekawa et al., 1993].

Yasuyuki Maekawa, Shoichiro Fukao, Mamoru Yamamoto, Manabu D. Yamanaka, Toshitaka Tsuda, Susumu Kato, R. F. W. First observation of the upper stratospheric vertical wind velocities using the Jicamarca VHF radar. Geophys. Res. Lett. 20, 2235–2238 (1993).

Balsley, B. B. & Gage, K. S. The MST radar technique: Potential for middle atmospheric studies. Pure Appl. Geophys. PAGEOPH 118, 452–493 (1980).

Kato, S. Validity on radar observation of middle- And upper-atmosphere dynamics. Earth, Planets Sp. 61, 545–549 (2009).

Woodman, R. F. and A. G. Radar Observations of Winds and Turbulence in the Stratosphere and Mesosphere. J. Atmos. Sci. 31, 493–505 (1974).

118: what does "integrated" mean in this context?

- We changed the wording to be more clear. We removed the word "integrated" and replaced it with running mean.

"The temperature profiles are calculated as a running mean over 2 h with a 15 min shift in time and binned to a vertical resolution of 1 km [Baumgarten2017]."

Figure 1 implies hard edges, but they're quite fuzzy. Not sure how to fix it this easily, but it might be worth a think. My own papers can be just as bad though on this...

- Thank you for the suggestion. We have added a sentence to the caption to make it clearer.

"Note that the response function for AIRS for vertical wavelengths below $\sim$26 km is very small (< 1\%). While this idealized figure implies hard cut-offs at certain vertical wavelengths, the nature of the filters (whether observational or imposed) is more of a gradual transition."

144: The transition from the case study in lines 139-144 to a more statistical study from line 144 onwards is very abrupt, and needs delineating more clearly

- We started a new paragraph to describe the statistical figure.

145: "as a function of time-averaged over an area" is ungrammatical

- It has been corrected.

Figure 2: a colourbar for the inset would be very useful

- Thank you for the comment. The colorbar has been added.

183: how long a temporal average? Could affect the results.

- It is a daily average. The clarification has been made.

"A daily average is applied, which is denoted by the over-bar above the temperatures. More details regarding lidar data processing can be found in [Baumgarten2017]."

206: they are *likely* to be related...

- Thank you for the suggestion. The sentence was changed based on the other reviewers comment.

"These events are beyond the scope of this paper because they are either generated by convective activity outside of the area we chose for this study or by other gravity wave sources, e.g., baroclinic instabilities."

208: that's a very precise height level - is it actually a pressure level? If so might be clearer to specify that, with the height approxn in brackets afterwards, eg something like "10hPa (~16km)". Same comment for line 209.

- We changed it to say around 40 km.